# Fast Recursive Computation of Sliding DHT with Arbitrary Step

**DOI:** 10.3390/s20195556

**Published:** 2020-09-28

**Authors:** Vitaly Kober

**Affiliations:** 1Department of Computer Science, CICESE, Ensenada 22860, Mexico; vkober@hotmail.com; 2Department of Mathematics, Chelyabinsk State University, Chelyabinsk 454001, Russia; 3Institute for Information Transmission Problems, Russian Academy of Sciences, Moscow 127051, Russia

**Keywords:** discrete Hartley transform, sliding algorithm, sensor noise removal, signal processing, short-time transform

## Abstract

Short-time (sliding) transform based on discrete Hartley transform (DHT) is often used to estimate the power spectrum of a quasi-stationary process such as speech, audio, radar, communication, and biomedical signals. Sliding transform calculates the transform coefficients of the signal in a fixed-size moving window. In order to speed up the spectral analysis of signals with slowly changing spectra, the window can slide along the signal with a step of more than one. A fast algorithm for computing the discrete Hartley transform in windows that are equidistant from each other is proposed. The algorithm is based on a second-order recursive relation between subsequent equidistant local transform spectra. The performance of the proposed algorithm with respect to computational complexity is compared with the performance of known fast Hartley transform and sliding algorithms.

## 1. Introduction

In today’s world, the study of real-life phenomena begins with sensors that convert the description of phenomena into digital signals. Since real sensors are sensitive to internal thermal, electronic, and environmental noise, the sensors’ output must be intelligently processed for subsequent use in a computer system. Short-time signal processing [1] is an appropriate technique for such a system, since it is capable of adaptively filtering long-term stationary (in wide-sense) and quasi-stationary signals. An example of quasi-stationary signal is speech. Frequency components of this signal change over time when the position of tongue and mouth changes, but on a short period of time they might be considered stable because tongue cannot move very quickly. Short-time processing is used in a variety of applications such as radar emitter recognition [2], heart sound classification [3], sensor noise removal [4], gearbox fault diagnosis [5], speech processing and spectral analysis [6], and adaptive linear filtering [7].

Basically, short-time transform is a time series of windowed signal transforms that can be used to perform time-frequency analysis. Local signal processing in a moving window (special case of short-time processing) can be carried out as follows: at each position of the running window, the coefficients of an orthogonal transform of the signal are modified to estimate only the central window element [8]. For time-varying signals, it is preferred that the window length be small enough so that the windowed signal would be approximately stationary. On the other hand, the window size should be large enough to provide a reasonable frequency resolution of the local signal. Typically, with local sliding processing, the window moves with a step of one along the signal. However, if it is necessary to process the signal at a high speed and the signal spectrum changes slowly over time, the window can move with a step of more than one. This approach has recently been proposed to design preview tools for multiresolution signal and image restoration [9].

Discrete Hartley transform (DHT) [10] is used in various practical applications such as speech spectral analysis [11], feature extraction and sea surface modeling [12], data compression [13], and signal interpolation [14]. Four types of DHT are classified [15,16]. Discrete Hartley and Fourier transforms of real signals and their relations with discrete sinusoidal transformations are discussed in detail [17,18]. Note that for real signals, all DHT coefficients can be used in the design of transform domain scalar filters. Representation of a signal in the DHT sliding domain is especially suitable for processing of time-varying, quasi-stationary data. Since calculating the DHT at each position of a running window is an intensive task, recursive algorithms can be used. First-order shift properties were obtained [19]. Since this approach is not very efficient in terms of computational complexity, four types of the sliding DHT were derived based on second-order recursive equations [20].

In this work, a recursive algorithm is proposed for fast computation of the DHT in a window sliding along the signal more than one sample at a time. The main contributions of the work are as follows:Second-order recursive equation between three consecutive equidistant DHT spectra is obtained using the z-transform technique.Efficient sliding DHT algorithm is proposed using the properties of discrete sinusoidal functions and the recursive equation.Computational complexity and running time of the proposed sliding algorithm are compared with the known sliding and fast DHT algorithms.

The paper is organized as follows. In Section 2, a second-order recursive equation between three consecutive equidistant DHT spectra is derived. In Section 3, a fast forward algorithm for computing the sliding DHT is suggested. Section 3 also presents a fast inverse sliding DHT algorithm. The performance of the proposed and common DHT algorithms in terms of computational complexity and runtime is given and discussed in Section 4. Finally, our conclusions are presented in Section 5.

## 2. Second-Order Equation for Recursive Computation of Sliding DHT

We recall here the definition of DHT. The following notations are used: $cs_{N}\left(rs\right)\equiv \cos\left(\frac{2\pi }{N}rs\right)$, $sn_{N}\left(rs\right)\equiv \sin\left(\frac{2\pi }{N}rs\right)$, $snc_{N}\left(rs\right)\equiv \frac{sn_{N}\left(rs\right)}{sn_{N}\left(s\right)}$
${\mathrm{cas}}_{N}\left(rs\right)\equiv \cos\left(\frac{2\pi }{N}rs\right)+\sin\left(\frac{2\pi }{N}rs\right)$, $\bar{{\mathrm{cas}}_{N}}\left(rs\right)\equiv \cos\left(\frac{2\pi }{N}rs\right)-\sin\left(\frac{2\pi }{N}rs\right)$, $W_{N}^{s}\equiv e^{i\frac{2\pi }{N}s}=cs_{N}(s)+isn_{N}(s),$ where the subscript *N* is the transform order, *r* is an integer, and s=0, 1,…N−1. For clarity, the normalization factor $1/\sqrt{N}$ is disregarded until the inverse transform.

Sliding DHT with a step of *p* can be defined as follows:(1)$$y_{s}\left(kp\right)=\sum_{n=-N_{1}}^{N_{2}}x\left(kp+n\right){\mathrm{cas}}_{N}\left(s\left(n+N_{1}\right)\right),$$
where {x(kp); *k* =…, −*N*_1_, −*N*_1_ + 1,…, 0, 1,…, *N*_2_, *N*_2_ + 1….} is a signal to be processed; $\left\{y_{s}\left(kp\right),\;s=0,\;1,\ldots N-1\right\}$ are the transform coefficients around time *kp*; *N* = *N*_1_ + *N*_2_ + 1 is the size of the moving window; *N*_1_ and *N*_2_ are arbitrary integer values.

A recursive relationship between three consecutive sliding spectra is given as [20]:(2)$$y_{s}\left(k+2\right)-2{\mathrm{cs}}_{N}\left(s\right)y_{s}(k+1)+y_{s}(k)=f_{s}(k),$$
where $f_{s}\left(k\right)=\left(x\left(k+N_{2}+2\right)-x\left(k-N_{1}+1\right)\right)\bar{cas_{N}}\left(s\right)+x\left(k-N_{1}\right)-x\left(k+N_{2}+1\right)$.

This is a linear inhomogeneous difference equation defined on *k*. For fixed *s*, (2) can be considered as a linear difference equation with constant coefficients. Linear time-invariant systems defined by such equations can be analyzed using the unilateral z-transform [1]. Suppose that we deal with a causal linear system (region of convergence is outside a circle on the z-plane) and nonzero initial conditions $y_{s}(0)$ and $y_{s}(1)$ are given. Applying the unilateral z-transform to (2) and using its shift property, we get:(3)$$z^{2}\left[Y_{s}\left(z\right)-y_{s}\left(0\right)-z^{-1}y_{s}\left(1\right)\right]-2{\mathrm{cs}}_{N}\left(s\right)z\left[Y_{s}\left(z\right)-y_{s}\left(0\right)\right]+Y_{s}\left(z\right)=F_{s}\left(z\right),$$
where $Y_{s}\left(z\right)$ and $F_{s}\left(z\right)$ are the z-transforms of $y_{s}(k)$ and $f_{s}(k)$, respectively.

We express $Y_{s}\left(z\right)$ as:(4)$$Y_{s}\left(z\right)=\frac{F_{s}\left(z\right)+y_{s}^{}\left(0\right)z\left(z-2{\mathrm{cs}}_{N}\left(s\right)\right)+y_{s}\left(1\right)z}{z^{2}-2{\mathrm{cs}}_{N}\left(s\right)z+1}.$$

For s≠0, there are two different roots of the denominator; that is, $q_{1}\left(s\right)=W_{N}^{s}$ and $q_{2}\left(s\right)=W_{N}^{-s}$. For s=0, the roots are equal to $q_{1}\left(s\right)=q_{2}\left(s\right)=1$. Let us carry out the partial fraction expansion of the equation:(5)$$D_{s}\left(z\right)=\frac{z}{z^{2}-2{\mathrm{cs}}_{N}\left(s\right)z+1}=\{\begin{array}{c} \frac{z^{-1}}{1-2{\mathrm{cs}}_{N}\left(s\right)z^{-1}+z^{-2}},s\neq 0\\ \frac{z^{-1}}{1-2z^{-1}+z^{-2}},s=0 \end{array},$$
and obtain
(6)$$D_{s}\left(z\right)=\{\begin{array}{c} \frac{1}{q_{1}\left(s\right)-q_{2}\left(s\right)}\left(\frac{1}{1-q_{1}{\left(s\right)}_{1}z^{-1}}-\frac{1}{1-q_{2}\left(s\right)z^{-1}}\right),s\neq 0\\ \frac{z^{-1}}{{\left(1-z^{-1}\right)}^{2}},s=0 \end{array}.$$

The inverse z-transform can be computed as follows [1]:(7)$$D_{s}\left(z\right)\overset{Z^{-1}}{\to }d_{s}\left(k\right)=\{\begin{array}{cc} \frac{q_{1}^{k}\left(s\right)-q_{2}^{k}\left(s\right)}{q_{1}\left(s\right)-q_{2}\left(s\right)}u\left(k\right)=\frac{sn_{N}\left(ks\right)}{sn_{N}\left(s\right)}u\left(k\right), & s\neq 0\\ ku(k), & s=0 \end{array},$$
where u(k) is the unit-step function defined as 1, for k≥0 and 0, for k<0.

Using (5), (4) can be represented as follows:(8)$$Y_{s}\left(z\right)=F_{s}\left(z\right)\left[z^{-1}D_{s}\left(z\right)\right]+y_{s}\left(0\right)\left[zD_{s}\left(z\right)-2{\mathrm{cs}}_{N}\left(s\right)D_{s}\left(z\right)\right]+y_{s}\left(1\right)D_{s}\left(z\right).$$

Using the shifting and convolution properties, the inverse transform is given as:(9)$$y_{s}(k)=\sum_{i=0}^{k}d_{s}\left(i-1\right)f_{s}\left(k-i\right)+y_{s}\left(0\right)\left[d_{s}\left(k+1\right)-2{\mathrm{cs}}_{N}\left(s\right)d_{s}\left(k\right)\right]+y_{s}\left(1\right)d_{s}\left(k\right).$$

Substituting (7) into (9) and taking into account that $d_{s}\left(k\right)=0$ for k<2,
$q_{1}\left(s\right)q_{1}\left(s\right)=1$ and $2{\mathrm{cs}}_{N}\left(s\right)=q_{1}\left(s\right)+q_{2}\left(s\right)$, we arrive at:(10)$$y_{s}(k)=\{\begin{array}{c} \sum_{r=1}^{k-1}f_{s}\left(k-r-1\right)r-y_{s}\left(0\right)\left(k-1\right)+y_{s}\left(1\right)k,\;s=0\\ \frac{\sum_{r=1}^{k-1}sn_{N}\left(rs\right)f_{s}\left(k-r-1\right)-y_{s}\left(0\right)sn_{N}\left(\left(k-1\right)s\right)+y_{s}\left(1\right)sn_{N}\left(ks\right)}{sn_{N}\left(s\right)},\;s=1,\ldots N-1 \end{array},$$
with k≥2.

Given two initial values $y_{s}(0)$ and $y_{s}(p)$
(p>1), one can obtain $y_{s}(1)$ from (10) as:(11)$$y_{s}(1)=\{\begin{array}{cc} \frac{y_{s}(p)+y_{s}(0)(p-1)-\sum_{r=1}^{p-1}f_{s}(p-r-1)r}{p}, & s=0\\ \frac{y_{s}(p)sn_{N}\left(s\right)+y(0)sn_{N}\left(\left(p-1\right)s\right)-\sum_{r=1}^{p-1}f(p-r-1)sn_{N}\left(rs\right)}{sn_{N}\left(ps\right)},\; & s=1,\ldots N-1 \end{array}.$$

Substituting (11) into (10), for arbitrary *k* we obtain:(12)$$y_{s}(k)=\{\begin{array}{cc} \frac{k}{p}y_{s}(p)+\frac{k-p}{p}\left(\sum_{r=1}^{p-1}f_{s}(r-1)r-y_{s}(0)\right)+\sum_{r=p}^{k-1}f_{s}(r-1)(k-r), & s=0\\ \frac{sn_{N}\left(\left(k-p\right)s\right)}{sn_{N}\left(ps\right)}\left(\sum_{r=1}^{p-1}f_{s}(r-1)\frac{sn_{N}\left(rs\right)}{sn_{N}\left(s\right)}-y_{s}(0)\right)+\frac{sn_{N}\left(ks\right)}{sn_{N}\left(ps\right)}y_{s}(p)+\sum_{r=p}^{k-1}f_{s}(r-1)\frac{sn_{N}\left(\left(k-r\right)s\right)}{sn_{N}\left(s\right)}, & s=1,\ldots N-1 \end{array}.$$

Finally, the DHT at the window position 2*p* is recursively computed using $y_{s}(0)$ and $y_{s}(p)$ as:(13)$$y_{s}(2p)=\{\begin{array}{cc} \sum_{r=1}^{p-1}\left(f_{s}(r-1)+f_{s}(2p-r-1)\right)r-y_{s}(0)+2y_{s}(p)+f_{s}(p-1)p, & s=0\\ \sum_{r=1}^{p-1}\left(f_{s}(r-1)+f_{s}(2p-r-1)\right)\frac{sn_{N}\left(rs\right)}{sn_{N}\left(s\right)}-y_{s}(0)+2cs_{N}\left(ps\right)y_{s}(p)+f_{s}(p-1)\frac{sn_{N}\left(ps\right)}{sn_{N}\left(s\right)}, & s=1,\ldots N-1 \end{array}.$$

This equation gives relationship between three consecutive equidistant DHT spectra, computed at times 0, p, and 2p.

## 3. Fast Sliding Algorithm for Computing DHT

### 3.1. Special Values of Discrete Sinusoidal Functions

Given below are special values (0, 1,−1) of discrete sinusoidal functions used in the performance analysis of the proposed sliding algorithm. Let us consider the following functions: $cs_{N}\left(rs\right)$, $snc_{N}\left(rs\right)$, and $\bar{cas_{N}}\left(s\right)$ with *s =* 0, 1*,…, N −* 1. The special values are shown in Table 1. Here, *r*, *l*, and *N* are arbitrary integer values, and the binary variable *b* takes the values {0, 1}. So, it is necessary to find such values of the variables *l* and *b* in order to obtain integer values of *s* in the range from 0 to *N* − 1.

The functions are used in the proposed algorithm in the operations of addition and multiplication; that is, when the functions are equal to 0, then the corresponding addition and multiplication operations can be discarded, and when they are equal to either −1 or 1, then the corresponding multiplication operations can be canceled. Therefore, the use of the special values can reduce the computational complexity of the algorithm.

### 3.2. Design of Forward Sliding Algorithm

Equation (13) for p>1 can be rewritten as:(14)$$y_{s}(2p)=\{\begin{array}{cc} \sum_{r=1}^{p}\left(B\left(r\right)+A\left(r\right)\right)r-y_{s}(0)+2y_{s}(p), & s=0\\ \bar{{\mathrm{cas}}_{N}}\left(s\right)\left(B\left(1\right)+\sum_{r=2}^{p}B\left(r\right)snc_{N}\left(rs\right)\right)+\left(A\left(1\right)+\sum_{r=2}^{p}A\left(r\right)snc_{N}\left(rs\right)\right)-y_{s}(0)+2cs_{N}\left(ps\right)y_{s}(p), & s=1,\ldots N-1 \end{array},$$
where {B(r)=Δ(r)+Δ(2p−r);A(r)=−Δ(r−1)−Δ(2p−r−1);
B(p)=Δ(p);
A(p)=−Δ(p−1);$\Delta \left(r\right)=x\left(r+N_{2}+1\right)-x\left(r-N_{1}\right);$r=1,…p−1}. At each window, the number of additions required for calculation of {B(r),A(r);r=1,…p} is 3p−2, since the coefficients {Δ(r);r=0,…p−1} have already been computed and stored at the position *p* of the window.

Suppose that *N* is odd. Let $g_{r}$, $h_{r}^{+}$, $h_{r}^{-}$ be the greatest common factors of N and *r, r +* 1*, r −* 1*,* respectively. The property of symmetry of the discrete function $snc_{N}\left(rs\right)$ is given as follows:$\{snc_{N}\left(rs\right)=snc_{N}\left(r\left(N-s\right)\right);\;s=1,\ldots \frac{N-1}{2},\;r=2,\ldots p\}.$ Let us compute $h_{r}=MAX\left(h_{r}^{+},h_{r}^{-}\right).$ For a fixed *r*, the quantities of 0 and ±1 of $\{snc_{N}\left(rs\right);\;s=1,\ldots \frac{N-1}{2},\;r=2,\ldots p\}$ are equal to $\frac{g_{r}-1}{2}$ and $\frac{h_{r}^{}-1}{2},$ respectively (see the second row of Table 1). So, the number of multiplications can be estimated as:(15)$$C_{MUL}=N-1+2\left(\left(p-1\right)\frac{N+1}{2}-\sum_{r=2}^{p}\frac{g_{r}+h_{r}^{}-2}{2}\right)+N=N\left(p+1\right)+3p-\sum_{r=2}^{p}\left(g_{r}+h_{r}^{}\right)-4.$$

Let us denote $SUM_{Q}\left(s\right)\equiv \sum_{r=1}^{p}Q\left(r\right)snc_{N}\left(rs\right)$. Using the symmetry property of the functions $\{SUM_{A}\left(s\right)=SUM_{A}\left(N-s\right),$$SUM_{B}\left(s\right)=SUM_{B}\left(N-s\right);s=1,\ldots \frac{N-1}{2}\},$ the number of additions can be estimated as:(16)$$C_{ADD}=2\left(\left(p-1\right)\frac{N+1}{2}-\sum_{r=2}^{p}\frac{g_{r}-1}{2}\right)+3N+3p-2=N\left(p+2\right)+5p-\sum_{r=2}^{p}g_{r}-4.$$

Additional expenses are required for calculating initial *p* coefficients.

Suppose that *N* is even. Equation (14) can be represented as follows:(17)$$y_{s}(2p)=\{\begin{array}{cc} \sum_{r=1}^{p}\left(B\left(r\right)+A\left(r\right)\right)r-y_{s}(0)+2y_{s}(p), & s=0\\ \sum_{r=1}^{p}\left(-B\left(r\right)+A\left(r\right)\right){\left(-1\right)}^{r-1}r-y_{s}(0)+2{\left(-1\right)}^{p}y_{s}(p), & s=\frac{N}{2}\\ \bar{{\mathrm{cas}}_{N}}\left(s\right)\left(B\left(1\right)+\sum_{r=2}^{p}B\left(r\right)snc_{N}\left(rs\right)\right)+\left(A\left(1\right)+\sum_{r=2}^{p}A\left(r\right)snc_{N}\left(rs\right)\right)-y_{s}(0)+2cs_{N}\left(ps\right)y_{s}(p), & s=1,\ldots ,N/2-1,N/2+1,\ldots ,N-1 \end{array}.$$

Let ${\bar{g}}_{r},$${\bar{h}}_{r}^{+}$, ${\bar{h}}_{r}^{-}$ be the greatest common factors of $\frac{N}{2}$ and *r, r +* 1*, r −* 1*,* respectively. In this case, the property of symmetry of the discrete function $snc_{N}\left(rs\right)$ is given as follows:$\{\overset{}{snc_{N}}\left(rs\right)=snc_{N}\left(r\left(N-s\right)\right)={\left(-1\right)}^{r-1}snc_{N}\left(r\left(\frac{N}{2}\pm s\right)\right),snc_{N}\left(r\frac{N}{2}\right)={\left(-1\right)}^{r-1}r\;;\;s=0,\ldots \left[\frac{N}{4}\right],\;r=2,\ldots p\},$ here [x/y] is the integer quotient. Let us compute ${\bar{h}}_{r}=MAX\left({\bar{h}}_{r}^{+},{\bar{h}}_{r}^{-}\right).$ For a fixed *r*, the quantities of 0 and ±1 of $\{\overset{}{snc_{N}}\left(rs\right);\;s=1,\ldots \left[\frac{N}{4}\right],\;r=2,\ldots p\}$ are equal to $\left[\frac{{\bar{g}}_{r}}{2}\right]$ and $\left[\frac{{\bar{h}}_{r}^{}}{2}\right],$ respectively (see second row of Table 1). The number of zeros of $\{cs_{N}\left(ps\right);\;s=1,\ldots N-1\}$ is equal to $2g_{p}$ (see the first row of Table 1). If N/8 is integer (see the third row of Table 1), then the number of zeros of the function $\{\bar{cas_{N}}\left(s\right);\;s=1,\ldots N-1\}$ equals $C_{cas}=2,$ otherwise $C_{cas}=0.$ It means that the zeros of the function can cancel the operations of multiplication and addition in the first term of (17) (see the last line of the equation) at the corresponding frequencies *s*. The number of ±1 of the function $\{\bar{cas_{N}}\left(s\right);\;s=0,\ldots N-1\}$ is equal to $2C_{cas}$. Finally, the number of multiplications can be estimated as follows:(18)$$\begin{array}{c} C_{MUL}=N-2+2\left(\left(p-1\right)\left(\left[\frac{N}{4}\right]+1-\frac{C_{cas}}{2}\right)-\sum_{r=2}^{p}\left(\left[\frac{{\bar{g}}_{r}}{2}\right]+\left[\frac{{\bar{h}}_{r}^{}}{2}\right]\right)\right)+N-2g_{p}-2C_{cas}=\\ 2\left(N+\left(p-1\right)\left[\frac{N}{4}\right]-\sum_{r=2}^{p}\left(\left[\frac{{\bar{g}}_{r}}{2}\right]+\left[\frac{{\bar{h}}_{r}^{}}{2}\right]\right)+p-g_{p}-2\right)-\left(p+1\right)C_{cas}. \end{array}$$

Using the symmetry property of the functions $\{SUM_{A}\left(s\right)=SUM_{A}\left(N-s\right),$$SUM_{B}\left(s\right)=SUM_{B}\left(N-s\right);s=1,\ldots \frac{N}{2}-1\},$ the number of additions can be estimated as:(19)$$\begin{array}{c} C_{ADD}=3N-1+2\left(\left(p-1\right)\left(\frac{N}{2}+1-\frac{C_{cas}}{2}\right)-\sum_{r=2}^{p}\left[\frac{{\bar{g}}_{r}}{2}\right]\right)-2g_{p}+3p-2=\\ N\left(p+2\right)-2\sum_{r=2}^{p}\left[\frac{{\bar{g}}_{r}}{2}\right]-2g_{p}+\left(p-1\right)\left(5-C_{cas}\right). \end{array}$$

Additional expenses are needed for calculating initial *p* coefficients.

To more clearly show the design of the algorithm, we provide an example for computing the sliding DHT coefficients. Assume that *N*_1_ = 7, *N*_2_ = 8, *N* = 16, *p* = 2, the DHT is computed at the window position *2p*
$\left\{y_{s}\left(2p\right),\;s=0,\;1,...15\right\}$. We borrow from the window position *p* two coefficients ${△}_{0}=x_{9}-x_{-7};\;{△}_{1}=x_{10}-x_{-6}$, and calculate the following auxiliary data:


${△}_{2}=x_{11}-x_{-5};{△}_{3}=x_{12}-x_{-4}$



$A_{1}=\Delta_{0}+\Delta_{2};\;B_{1}=\Delta_{1}+\Delta_{3}$



$S_{0}=2\Delta_{2};S_{1}=1.84776\Delta_{2};S_{2}=1.41421\Delta_{2};S_{3}=0.76537\Delta_{2}$



$C_{0}=2\Delta_{1};C_{1}=1.84776\Delta_{1};C_{2}=1.41421\Delta_{1};C_{3}=0.76537\Delta_{1}$



$Q_{0}^{\pm }=A_{1}\pm C_{0};Q_{1}^{\pm }=A_{1}\pm C_{1};Q_{2}^{\pm }=A_{1}\pm C_{2};Q_{3}^{\pm }=A_{1}\pm C_{3}$



$M_{0}^{\pm }=B_{1}\pm S_{0};M_{1}^{\pm }=B_{1}\pm S_{1};M_{2}=B_{1}-S_{2};M_{3}^{\pm }=B_{1}\pm S_{3}$



$F_{1}=0.5412M_{1}^{+};F_{3}=-0.5412M_{3}^{+};F_{5}=-1.30656M_{3}^{-}$


$F_{6}=-1.41421M_{2};F_{7}=-1.30656M_{1}^{-}$.

DHT coefficients are computed as follows:


$y_{0}\left(2p\right)=M_{0}^{+}-Q_{0}^{+}-y_{0}\left(0\right)+2y_{0}\left(p\right)$



$y_{1}\left(2p\right)=F_{1}-Q_{1}^{+}-y_{1}\left(0\right)+1.41421y_{1}\left(p\right)$



$y_{2}\left(2p\right)=-Q_{2}^{+}-y_{2}\left(0\right)$



$y_{3}\left(2p\right)=F_{3}-Q_{3}^{+}-y_{3}\left(0\right)-1.41421y_{3}\left(p\right)$



$y_{4}\left(2p\right)=-B_{1}-A_{1}-y_{4}\left(0\right)-2y_{4}\left(p\right)$



$y_{5}\left(2p\right)=F_{5}-Q_{3}^{-}-y_{5}\left(0\right)-1.41421y_{5}\left(p\right)$



$y_{6}\left(2p\right)=F_{6}-Q_{2}^{-}-y_{6}\left(0\right)$



$y_{7}\left(2p\right)=F_{7}-Q_{1}^{-}-y_{7}\left(0\right)+1.41421y_{7}\left(p\right)$



$y_{8}\left(2p\right)=-M_{0}^{-}-Q_{0}^{-}-y_{8}\left(0\right)+2y_{8}\left(p\right)$



$y_{9}\left(2p\right)=-F_{1}-Q_{1}^{-}-y_{9}\left(0\right)+1.41421y_{9}\left(p\right)$



$y_{10}\left(2p\right)=-Q_{2}^{-}-y_{10}\left(0\right)$



$y_{11}\left(2p\right)=-F_{3}-Q_{3}^{-}-y_{11}\left(0\right)-1.41421y_{11}\left(p\right)$



$y_{12}\left(2p\right)=B_{1}-A_{1}-y_{12}\left(0\right)-2y_{12}\left(p\right)$



$y_{13}\left(2p\right)=-F_{5}-Q_{3}^{+}-y_{13}\left(0\right)-1.41421y_{13}\left(p\right)$



$y_{14}\left(2p\right)=-F_{6}-Q_{2}^{+}-y_{14}\left(0\right)$


$y_{15}\left(2p\right)=-F_{7}-Q_{1}^{+}-y_{15}\left(0\right)+1.41421y_{15}\left(p\right)$.

The computational complexity of the algorithm is 25 multiplications and 61 additions.

### 3.3. Design of Inverse Sliding Algorithm

The inverse sliding DHT computes only the element x(kp) of the window. The inverse algorithm can be written as follows:(20)$$x\left(kp\right)=\frac{1}{N}\sum_{s=0}^{N-1}y_{s}\left(kp\right)cas_{N}\left(N_{1}s\right).$$
where $N=N_{1}+N_{2}+1$. 

If x(kp) is the central window element, that is, $N=2N_{1}$, then one can simplify the inverse transform to:(21)$$x\left(kp\right)=\frac{1}{N}\sum_{s=0}^{N-1}y_{s}\left(kp\right){\left(-1\right)}^{s}.$$

Finally, for $N_{1}=0$ the inverse transform is given as:(22)$$x\left(kp\right)=\frac{1}{N}\sum_{s=0}^{N-1}y_{s}\left(kp\right).$$

The proposed algorithms require only one multiplication and *N* − 1 additions.

## 4. Results and Discussion

In this section, using computer simulation, we analyze the performance of the proposed algorithm with respect to the computational complexity and execution time and compare it with that of fast DHT and conventional recursive algorithms. Among fast DHT algorithms, the most popular are fast radix-2 [21,22]. The recursive sliding (with a step of one) DHT algorithms [19,20] are carried out *p* times for computing the DHT spectra at equidistant positions *kp* of the window. Table 2 and Table 3 show the computational complexity in terms of multiplications and additions, respectively, of the proposed, fast radix-2 DHT, and known sliding algorithms at a fixed window position for *p* = 2 when *N* varies.

It can be seen that the proposed algorithm for *p* > 2 outperforms the conventional sliding DHT algorithms. Note that the proposed algorithm is more efficient than the fast and conventional recursive algorithms when the window length increases.

In modern processors, the execution times of floating point multiplication and addition are comparable. So, further we will estimate the computational complexity with respect to flop counts (real additions and multiplications). Table 4 shows the computational complexity of the tested algorithms for *N* = 256 when *p* varies.

It can be seen that the proposed algorithm is more efficient than the fast DHT algorithm when the step p<5. This boundary value of the step, when the proposed algorithm is still better than the fast DHT algorithm, increases with increasing the window length.

Obviously, the execution time of any algorithm depends on the characteristics of a computer used in a particular implementation of the algorithm. Now, with the help of computer simulation, we want to illustrate how the theoretical computational complexity of the tested algorithms relates to the execution time of the implemented algorithms. Computer simulation was performed on a laptop with an Intel Core i7-2630QM processor with 8 GB of RAM using the MATLAB R2012b. Experiments were carried out 100 times to ensure statistically correct results. Average time results for each algorithm were calculated. Figure 1 shows the performance of the tested algorithms in terms of runtime: DHT is the discrete Hartley transform given by definition, Fast DHT is implemented in Matlab, SDHT is sliding DHT [20], ALG is the proposed algorithm.

It can be seen that the obtained results are in good agreement with the results in Table 4. Only the performance of the fast algorithm implemented in the MATLAB is slightly lower than that of the algorithm [22].

There are four types of DHT [16] that are suitable for the time varying processing of different signal models. In this paper, the second order recursive equation and fast recursive algorithm have been proposed for only one type of discrete Hartley transform (DHT-I). In the future, the same approach can be used to derive recursive equations and design fast recursive algorithms for other types of DHT, which will allow the system to work effectively with various signal models.

## 5. Conclusions

A second-order recursive equation between three consecutive equidistant DHT spectra was obtained utilizing of the unilateral z-transform technique. Using the properties of discrete sinusoidal functions and the recursive equation, a fast sliding DHT algorithm was proposed. A fast inverse sliding DHT transform was also presented. The computational complexity of the proposed sliding algorithm was compared with the known running and fast DHT algorithms. The proposed algorithm was implemented on a laptop, and it was shown that the theoretical computational complexity and execution time of the implemented algorithm are in good agreement.

## Figures and Tables

**Figure 1 sensors-20-05556-f001:**
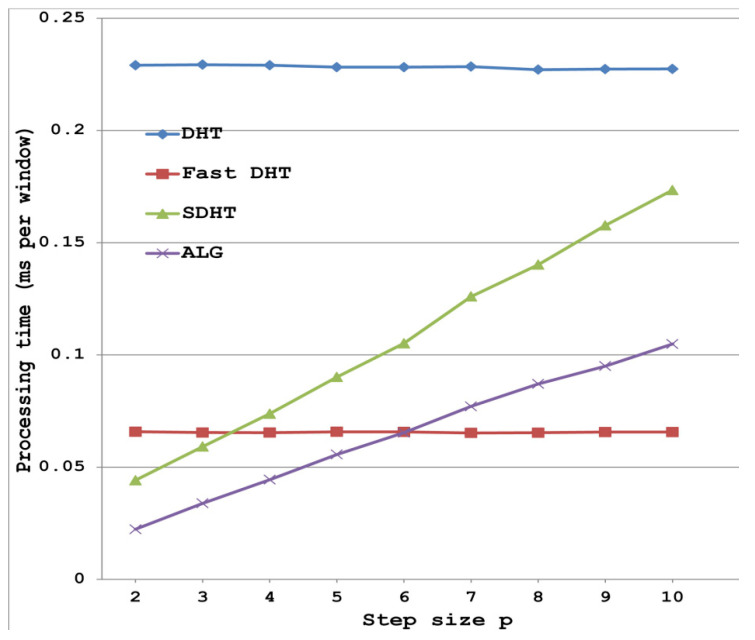
Performance of the tested algorithms in terms of runtime (milliseconds) per window when *N* = 256 and *p* varies from 2 to 10.

**Table 1 sensors-20-05556-t001:** Special values of the discrete sinusoidal functions.

Functions	Values		
	0	1	−1
$cs_{N}\left(rs\right)$	$s=\frac{N}{4r}\left(2l+1\right)$	$s=\frac{N}{r}l$	$s=\frac{N}{2r}\left(2l+1\right)$
$snc_{N}\left(rs\right)$	$s=\frac{N}{2r}l;s\neq 0,\frac{N}{2}$	$s=\frac{N}{2}\frac{\left(2l+b\right)}{\left(r+2b-1\right)}$	$s=\frac{N}{2}\frac{\left(2l+b\right)}{\left(r-2b+1\right)}$
$\bar{cas_{N}}\left(s\right)$	$s=\frac{N}{8}\left(4l+1\right)$	$s=0;s=\frac{3N}{4}$	$s=\frac{N}{4};s=\frac{N}{2}$

**Table 2 sensors-20-05556-t002:** Comparison of the tested algorithms with respect to multiplications for computing sliding DHT with *p* = 2.

Algorithms	Number of Operations, *N* = 2*^M^*	*N*-Length of Sliding Window					
		16	32	64	128	256	512
Fast DHT [22]	(*M* − 3)*N*/2 + 2	10	34	98	258	642	1538
Ref. [19]	(2*N* − 1)*p*	62	126	254	510	1022	2046
Ref. [20]	(5/4(*N* − 4)−1)*p*	28	68	148	308	628	1268
Proposed ALG	Equation (18)	25	68	148	308	628	1268

**Table 3 sensors-20-05556-t003:** Comparison of the tested algorithms with respect to additions for computing sliding DHT with *p* = 2.

Algorithms	Number of Operations, *N = 2^M^*	*N*-Length of Sliding Window					
		16	32	64	128	256	512
Fast DHT [22]	(*M* + 9)*N*/2-*M*^2^ − b3*M* − 6	70	178	420	948	2082	4494
Ref. [19]	(3*N* − 1)*p*	94	190	382	766	1534	3070
Ref. [20]	(5*N*/2 + 2)*p*	84	164	324	644	1284	2564
Proposed ALG	Equation (19)	61	125	253	509	1021	2045

**Table 4 sensors-20-05556-t004:** Comparison of the tested algorithms in terms of flops for computing sliding DHT, *N* = 256.

Algorithms	*p*-Step of Sliding Window				
	2	3	4	5	6
Fast DHT [22]	2724	2724	2724	2724	2724
Ref. [19]	2556	3834	5112	6390	7668
Ref. [20]	1912	2868	3824	4780	5736
Proposed ALG	1649	2036	2403	2798	3177

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
