# Peer review of "Fast Recursive Computation of Sliding DHT with Arbitrary Step"

_sensors, 2020, doi:10.3390/s20195556_

Round 1
Reviewer 1 Report
In equation 17 the condition s=1,...,N-1 includes s=N/2
Punctuation formatting should be improved
Author Response
- In equation 17 the condition s=1,...,N-1 includes s=N/2
Thank you, I corrected the condition by excluding S=N/2.
- Punctuation formatting should be improved
I improved the punctuation usage throughout the paper.
Reviewer 2 Report
The paper presents a novel implementation of the sliding Discrete Hartley Transform with an arbitrary step
- Please define the context of the "short-time signal processing" term. Does the author refer to algorithms that use a relatively small set of samples?
- In lines 82-86, the term "rs" (inside of the brackets) is introduced. What does it mean?
- Please, include a full comparison between the time-frequency spectra obtained with the proposal, the short-time Fourier transform, and the continuous wavelet transform. The author mentions that DHT can be used for spectral analysis. An appropriate test signal can be a chirp one.
Author Response
Responses to your comments are provided in the attached file.

Reviewer 3 Report
Dear Author,
The paper is well written and has good logical flow. The English language of the paper is good and text is easy to read and understand.
Author have proposed low complexity (compared to provided references) algorithm for for fast computing the DHT in windows that are equidistant from each other.
What needs to be done to have decreased processing time for proposed algorithm for step size p larger than 6 (see Figure 1)?
Thank you in advance!
Best regards,
Reviewer
Author Response
1. What needs to be done to have decreased processing time for proposed algorithm for step size p larger than 6 (see Figure 1)?
The processing time of the proposed algorithm can be reduced either by increasing the window size, or by using parallel processing by several sliding algorithms with shifted initial processing times.
Reviewer 4 Report
The author has done an interesting work. In this paper, a recursive algorithm is proposed for fast computing the DHT in windows that are equidistant from each other. Compared with the classic algorithms, the proposed algorithm has shown excellent performances in terms of computational complexity and runtime. Besides, the significance of content is high, and the English writing is good. So I would recommend this paper for publication.
Author Response
Thank you.
Reviewer 5 Report
I found the work quite consistent and well written. I only found a small typo, or at least I think it is a typo, in eq. 9, last component before the end bracket is r(k), and I think it should be d_s(k). Although not really revolutionary, I think the idea is very interesting, original and mathematically consistent, and for some researchers involved in the use of these transforms it can be of interest.
Author Response
I found the work quite consistent and well written. I only found a small typo, or at least I think it is a typo, in eq. 9, last component before the end bracket is r(k), and I think it should be d_s(k).
Thank you, I corrected the typo.
All changes are marked with yellow color in the paper.
Round 2
Reviewer 2 Report
The author has addressed all the concerns inquired by this reviewer.
Author Response
I submitted the response in the attached pdf file.
Best regards,
V. Kober
